# Assessment of the Nutrient Removal Potential of Floating Native and Exotic Aquatic Macrophytes Cultured in Swine Manure Wastewater

**DOI:** 10.3390/ijerph17031103

**Published:** 2020-02-10

**Authors:** Lei Xu, Siyu Cheng, Ping Zhuang, Dongsheng Xie, Shiyu Li, Dongming Liu, Zhian Li, Faguo Wang, Fuwu Xing

**Affiliations:** 1South China Botanical Garden, Chinese Academy of Sciences, Guangzhou 510650, China; xulei@scbg.ac.cn (L.X.); zhuangp@scbg.ac.cn (P.Z.); xiedongsheng@scbg.ac.cn (D.X.); lishiyu@scbg.ac.cn (S.L.); liudm@scib.ac.cn (D.L.); lizan@scbg.ac.cn (Z.L.); xinfw@scbg.ac.cn (F.X.); 2College of Life Sciences, University of Chinese Academy of Sciences, Beijing 10049, China; 3College of Geography, Sun Yat-sen University, Guangzhou 510275, China; chensy540@163.com; 4Southern Marine Science and Engineering Guangdong Laboratory (Guangzhou), South China Botanical Garden, Chinese Academy of Sciences, Guangzhou 511458, China; 5College of Life Sciences, Zhongkai University of Agriculture and Engineering, Guangzhou 510225, China; 6Key Laboratory of Plant Resources Conservation and Sustainable Utilization, Guangdong Provincial Key Laboratory of Applied Botany, South China Botanical Garden, Chinese Academy of Sciences, Guangzhou 510650, China

**Keywords:** eutrophication, nutrient removal, floating macrophytes, water purification

## Abstract

Although eutrophication and biological invasion have caused serious harm to aquatic ecosystems, exotic and even invasive plants have been used extensively in phytoremediation water systems in China. To identify native aquatic plants with excellent water restoration potential, two representative native floating aquatic plants from Guangdong Province, namely *Ludwigia adscendens* (PL) and *Trapa natans* (PT), were selected, with *Eichhornia crassipes* as a control, to study their growth status, adaptability, and nutrient removal potentials in swine manure wastewater. The results demonstrated that the two native plants offered greater advantages than *E. crassipes* in water restoration. Within 60 days, PL and PT exhibited excellent growth statuses, and their net biomass growth rates were 539.8% and 385.9%, respectively, but the *E. crassipes* decayed and died with an increasing HRT (hydraulic retention time). The PL and PT could adjust the pH of the wastewater, improve the dissolved oxygen and oxidation-reduction potential, and reduce the electrical conductivity value. The removal rates of NH_4_^+^–N, NO_3_^−^–N, NO_2_^−^–N, total nitrogen, total phosphorus, chemical oxygen demand (COD), and Chl-*a* in the PL group reached 98.67%, 64.83%, 26.35%, 79.30%, 95.90%, 69.62%, and 92.23%, respectively; those in the PT group reached 99.47%, 95.83%, 85.17%, 83.73%, 88.72%, 75.06%, and 91.55%, respectively. The absorption contribution rates of total nitrogen (TN) and total phosphorus (TP) in the PL group were 40.6% and 43.5%, respectively, while those in the PT group were 36.9% and 34.5%, respectively. The results indicated that *L. adscendens* and *T. natans* are both promising aquatic plants for application to the restoration of swine manure wastewater in subtropical areas.

## 1. Introduction

Eutrophication and biological invasion are widespread problems in rivers, lakes, and coastal oceans, caused by over enrichment with nitrogen and phosphorus and invasion of alien organisms [1,2], which seriously degrades aquatic ecosystems and impairs the use of water for drinking, industry, agriculture, recreation, and other purposes [3]. Many water bodies are experiencing an abrupt change from submerged macrophytes to phytoplankton domination, and a sharp decrease in native aquatic biodiversity [4,5], which is a catastrophe for aquatic ecosystems. Pollutants enter water bodies through surface runoff, erosion, rainfall, and groundwater [6]. With the rapid development of aquaculture all over the world, the runoff from animal farms has become a huge source of water pollution, where excessive nutrients are generated to accelerate eutrophication [7]. To address these serious issues, many conventional and novel methods using physical, chemical, and biological processes have been applied to treat animal wastewater over the past several decades [8,9]. Among these methods, phytoremediation using aquatic plants with high productivity and nutrient removal capability has received increasing public attention in recent years [10,11].

In tropical and subtropical regions, free-floating macrophytes are the dominant vegetation used in wastewater restoration [4]. Numerous studies have documented their ability to remediate eutrophic water bodies [12], including the following positive attributes: (1) high productivity, (2) high nutritive value, and (3) ease of stocking and harvesting [13]. However, most previous studies focused on the remediation efficiency of municipal or domestic sewage by exotic plant species [14], but ignored the study on aquaculture wastewater and the exploration and application of native aquatic plants. For example, the invasive floating aquatic plants *Eichhornia crassipes* and *Pistia stratiotes* have been used extensively in phytoremediation water systems [15,16], which poses a significant threat to biodiversity and global ecosystem stability regarding their possible removal from the remediation systems and placement in natural water bodies. It is generally accepted that, compared to many native macrophytes, these exotic or invasive plants exhibit a substantially higher nutrient removal efficiency, owing to their greater nutrient uptake capacity, higher growth rate, and greater biomass production [17]. This is because substantial time and energy have been focused on the study of exotic aquatic plants, but few efforts have been directed at identifying native plants with both ornamental value and water restoration potential. As reported by Zhang [18], the N and P removal abilities of the native plants *Ludwigia peploides* subsp. *stipulacea* in Taihu Lake, China, were 2.9 and 0.7 times those of *E. crassipes*, respectively, which could replace certain exotic plants to purify water bodies in practical engineering. The native species adapted to the local environment conditions effectively, with high ecological security and cultural value. It is significant to explore, discover, and apply native aquatic plants for the reconstruction and remediation of water systems, which can reduce the invasive harm caused by exotic plants in the environment.

The objective of this study was to compare the potential of native and invasive floating aquatic macrophytes in the restoration of swine manure wastewater. Prior to the experiment, we had conducted a comprehensive survey of aquatic plant resources in Guangdong Province [19], and based on their living habitats, population statuses, landscape characteristics, and potential utilization values, some local aquatic plants were selected for indoor pre-experiments to discover the right plants. Due to the severe damage caused by the invasive plant *E. crassipes* to Guangdong Province, China, two representative native floating aquatic plants, *Ludwigia adscendens* and *Trapa natans,* were selected to study their nutrient removal potential in swine manure wastewater. The results of this study may provide a reference for engineering practice, and contribute to the construction of ecological civilization in China.

## 2. Materials and Methods

### 2.1. Experimental Design

All experiments were conducted in a greenhouse located in Sihui, Guangdong Province, South China, from May 1 to July 1, 2018. The average day and night temperatures were 33 and 25 °C, respectively. The average sunshine duration was 13 h. The initial water quality parameters were as follows: temperature (T, 29.15 ± 0.15 °C); pH (7.84 ± 0.09); dissolved oxygen (DO, 1.49 ± 1.09 mg/L); oxidation-reduction potential (ORP, 110 ± 12 mV); electrical conductivity (EC, 434 uS/cm); total nitrogen (TN, 21.60 ± 2.60 mg/L); ammonium (NH_4_^+^, 7.84 ± 1.10 mg/L); nitrate (NO_3_^−^, 0.06 ± 0.02 mg/L); NO_2_^−^ (0.03 ± 0.01 mg/L); total phosphorus (TP, 5.50 ± 1.00 mg/L); chemical oxygen demand (COD, 177 ± 77.00 mg/L); and chlorophyll *a* (Chl-*a*, 413.00 ± 61.80 mg/L).

A total of 12 rectangular plastic tanks, measuring 60 cm in length, 50 cm in width, and 30 cm in height, were used for the experiments with 75 L of swine manure wastewater and 200 g of aquatic plants, as illustrated in Figure 1. The tanks were divided into four groups: (i) *E. crassipes* (PE), (ii) *L. adscendens* (PL), (iii) *T. natans* (PT), and (iv) black control (BC; no plants). Each group had triplicate tanks. All plants were collected from the local river of Sihui and were grown in running water for one week prior to the start of the experiment.

Wastewater samples were collected from each tank at various intervals (typically 0, 15, 30, 45, and 60 days), by dipping a 500 mL graduated cylinder at three locations across the container surface and combining these during mid-morning. Care was taken to minimize disturbance of the plants. The aliquots of each sample were filtered, and both the filtered and unfiltered portions were immediately stored at 4 °C. The parameters measured included T, pH, DO, ORP, EC, TN, NH_4_^+^–N, NO_3_^−^–N, NO_2_^−^–N, TP, Chl-*a*, and COD. Losses in the culture volume owing to evapotranspiration were countered by the addition of deionized water to the original level every second day. Water sampling was performed on the second day following the volume adjustment, so that the deionized water additions had a minimal impact on the measurements.

Aquatic plants were sampled from the treatment plots. The roots, shoots, and fruit of the tested plants were separated, washed three times with deionized water, briefly treated at 105 °C, and then dried to a constant weight at 65 °C. The dried plant material was ground, and the N and P contents were measured. At the end of the experiment, all plants were removed from their tanks and weighed.

In addition to the regular sampling, the plants were also periodically harvested to maintain optimal plant density. In each harvest, the total fresh weight of the plants was recorded, the plant moisture was determined, and the total quantity of dry plant biomass yield was calculated for each plot. The total amounts of N and P removed from the water by the harvested plants were quantified by multiplying the amounts of plant biomass by the N and P concentrations in the plant.

### 2.2. Chemical Analysis

The TN was determined by a TOC analyzer (TOC-L, SHIMADZU (Hong Kong) Limited, Kyoto, Japan). The NH_4_^+^–N, NO_3_^−^–N, and NO_2_^−^–N were determined by a flow injection analyzer (QC8000, LACHAT, Milwaukee, WI, USA). The TP was determined using the ammonium molybdate spectrophotometric method (GB11893-1989). The COD was determined using a spectrophotometer (DRB 200, Hach, Loveland, CO, USA). The Chl-*a* was determined spectrophotometrically (HJ 897-2017). The other physical and chemical characteristics, including T, pH, ORP, DO, and EC, were obtained using a portable multimeter (EXO2, YSI, Ohio, OH, USA).

The plant N concentration was determined using a CN analyzer (vario Max CN, Elemental Analysensystem GmbH, Hanau, Germany). Sub-samples (each 0.400 g) of the plant material were digested with 5 mL of concentrated HNO_3_ in a digestion tube using a block digestion system (AIM 500-C, A.I. Scientific Inc., Hornibrook, Australia), and the P concentration in the digester was determined using inductively coupled plasma atomic emission spectroscopy.

### 2.3. Data Analysis

In this study, all data were statistically analyzed by one-way ANOVA using the R-3.4.3 software (https://cran.r-project.org/bin/windows/base/old/3.4.3/), and significant differences were tested using the least significant difference and Duncan multiple comparisons (*p* = 0.05). In this case, *R*_1_ represents the total removal rate of nutrients in the wastewater; *R*_2_ represents the absorption contribution rate of the plants for the nutrient removal in the wastewater; and *R*_3_ represents the relative growth rate of the plant biomass in each group. The computation formulae are expressed as follows:*R*_1_ = (*C_i_* − *C_r_*)/*C_i_* × 100%(1)
*R*_2_ = (*P_f_* − *P_i_*)/(75*C_i_* − 75*C_r_*) × 100%(2)
*R*_3_ = (*W_f_* − *W_i_*)/*W_i_* × 100%(3)
where *C_i_* (mg/L) represents the initial concentration of nutrients in each tank; *C_r_* (mg/L) represents the residual concentration; *P_i_* and *P_f_* (mg) represent the initial and final N or P contents of the aquatic macrophytes, respectively; and *W_i_* and *W_f_* (mg) represent the initial and final fresh weights of the plants, respectively.

## 3. Results

### 3.1. Visual Observations and Biomass Production

All of the plants selected for the experiment were collected locally and had few roots at the beginning. We selected young and thriving individuals. During the test, as illustrated in Figure 2, the *L. adscendens* grew vigorously and its biomass increased significantly. Its floating stems grew rapidly, from 15 to 200 cm, with numerous branched, new creeping, rooting, and white spindle-shaped pneumatophores in clusters at the nodes. Moreover, as illustrated in Figure 3, *T. natans* was rapidly established in the wastewater, with the surface area of the containers being completely covered. Its leaves and roots grew rapidly; the diameter of its leaves reached at least twice the initial diameter, at approximately 20 to 25 cm; and the length of its root system reached 0.5 m, at least 20 times that of the initial length. The leaves and roots of the *E. crassipes* grew normally, and the biomass nearly doubled on the 15th day of the experiment. Thereafter, an abnormality appeared: from the 20th day, the stems, leaves, and roots gradually whitened and further decayed and died, as illustrated in Figure 4. The net biomass growth rates of the PL, PT, and PE at the end of the experiment are presented in Table 1, which indicates that the two native aquatic plants exhibited higher growth rates and superior adaptability in the swine manure wastewater compared to the *E. crassipes*.

### 3.2. Nutrient Concentrations and Absorption Capacities of the Plants

During the experiment, the TN and TP concentrations in the *L. adscendens* and *T. natans* plant tissues all decreased continuously with an increasing HRT (hydraulic retention time) (Table 2 and Table 3). The *E. crassipes* is not discussed here because of its death. The absorption contribution rate of plants (*R*_2_) refers to the proportion of N and P contents removed by the plants through absorption to the total amount removed in the water. In this experiment, the *R*_2_ values of the TN in the PL and PT groups were 40.6% and 36.9%, respectively; the *R*_2_ values of the TP were 43.5% and 34.5%, respectively. The results indicated that *L. adscendens* and *T. natans* exhibited a significant enrichment ability for N and P in the swine manure wastewater.

### 3.3. Variations in pH, DO, ORP, and EC in Wastewater

The values of the pH, DO, ORP, and EC in the wastewater with an increasing HRT are illustrated in Figure 5. The initial pH value of the tested water was 7.71 ± 0.19, which is weakly alkaline. During the experiment, the pH values in the PL and PT groups were reduced to neutral on the 15th day, and then remained stable; the pH values in the PE and BC groups fluctuated significantly and eventually reached 8.

The *C_i_* value of the DO in the wastewater was 1.46 ± 1.12 mg/L. During the first 15 days, the DO concentrations in the three plant groups all increased rapidly, while that in the BC group decreased, indicating that the aquatic plants could improve the DO concentration in the wastewater. On the 60th day of the experiment, the DO concentrations in the four groups all exceeded 5 mg/L, with the exception of PT (3.59 ± 0.07 mg/L). The change trend of the ORP exhibited a similar pattern to that of the DO. At the end, the average ORP values in the PL, PE, BC, and PT groups were 307.03, 279.70, 277.57, and 203.00 mV, respectively, all substantially higher than the initial value (120.11 ± 6.02 mV).

The initial EC value in the wastewater was 432.0 ± 2.0 uS/cm. The EC values of all treatments decreased and the water quality improved with an increasing HRT. At the end of the experiment, the EC values of the PL, PT, BC, and PE groups decreased to 79.8, 123.4, 263.2, and 304.3 uS/cm, respectively, which demonstrated that the two native floating aquatic plants were effective at reducing the EC value in the wastewater.

### 3.4. Nutrient Reduction in Wastewater

As a major nutrient for aquatic ecology, excess N can lead to eutrophication of surface waters [14]. The concentration and removal efficiencies of NH_4_^+^, NO_3_^−^, and NO_2_^−^, varying with an increasing HRT during the experiment, are illustrated in Figure 6.

The *C_i_* value of the NH_4_^+^ was 7.83 ± 0.36 mg/L, which decreased rapidly with an increasing HRT. On the 15th day of the experiment, the *R*_1_ values in the PT, PE, PL, and BC groups were 94.98%, 92.59%, 91.61%, and 78.75%, respectively. On the 60th day, the *C_r_* values of the NH_4_^+^ in the four groups were all less than 0.1 mg/L (*R*_1_ > 98%). During the experiment, the *R*_1_ value of the NH_4_^+^ in the BC group was high, but lower than that of the plant groups at the beginning, and even reached the same level as the plant groups in the end, although the change trend was slower.

The *C_i_* value of the NO_3_^−^ was 0.073 ± 0.018 mg/L. On the 15th day of the experiment, the *R*_1_ values of the NO_3_^−^ in the PT, BC, and PL groups were 57.95%, 40.43%, and 38.91%, while the *C_r_* values were 0.055, 0.030, and 0.035 mg/L, respectively. Moreover, the *C_r_* of the NO_3_^−^ in the PE group increased to 9.85 mg/L and reached almost 13,715 times the initial value.

The *C_i_* value of the NO_2_^−^ was 0.026 ± 0.0017 mg/L. During the entire experiment, the *R*_1_ values of the NO_2_^−^ in the four groups were not high, and some even exhibited negative values. On the 15th day, the *R*_1_ values of the PT, PL, and BC groups were 75.37%, 62.04%, and 22.82%, respectively, while the *C_r_* values were 0.0062, 0.011, and 0.020 mg/L, respectively. Furthermore, the *C_r_* in the PE increased to 6.82 mg/L and reached nearly 26,876 times the initial value, which was similar to the change trend of the NO_3_^−^ in the wastewater. At the end of the experiment, the *C_i_* values of the NO_2_^−^ in all groups remained at low levels.

The *C_i_* value of the TN was 23.07 ± 1.15 mg/L. During the first 15 days of the experiment, the TN concentrations of the PT and PL groups decreased rapidly (*C_r_* = 5.89 ± 0.60; *C_r_* = 7.73 ± 0.18 mg L^−1^), and the *R*_1_ values of the TN were 74.80% and 68.05%, respectively—significantly higher than those of the BC (43.45%) and PE (27.37%) groups (*p* < 0.05). Thereafter, the *R*_1_ values of the TN in PT and PL varied little with an increasing HRT, but those in BC and PE increased rapidly. At the end of the experiment, the *R*_1_ values of the TN in the PT, PL, PE, and BC groups were 84.05%, 79.04%, 77.84%, and 72.98%, respectively, while the *C_r_* values of the TN were 3.80, 5.01, 5.51, and 5.80 mg/L, respectively.

P is an important nutrient in the composition of biological life, and a highly significant limiting factor for water eutrophication [14]. In this experiment, the *C_i_* value of the TP was 5.39 ± 0.66 mg/L. The variations in the TP concentration and the homologous removal rate are illustrated in Figure 7. During the first 15 days, the *C_r_* values in the PL and PT groups decreased rapidly (*C_r_* = 0.79 ± 0.03; *C_r_* = 2.92 ± 0.08 mg L^−1^), and the *R*_1_ values of the TP were 85.11% and 51.66%, respectively, which were significantly higher than those of the BC (13.20%) and PE (12.77%) groups (*p* < 0.05). Subsequently, the *R*_1_ values of the TP in the PL, PT, and BC groups all increased slightly with an increasing HRT, while that in PE began to decrease from the 45th day. At the end, the *R*_1_ values of the TP in the PL, PT, BC, and PE groups were 95.90%, 88.72%, 38.32%, and 26.82%, respectively. The results demonstrated that the two native floating aquatic plants exhibited significant advantages in TP removal compared to the blank control and *E. crassipes*.

### 3.5. Chl-*a* and COD Reduction

In this experiment, the removal efficiency and concentration of the COD and Chl-*a* exhibited a similar pattern to that of the nutrients, which can be observed in Figure 8 and Figure 9, respectively. The most notable difference was in the BC group, where the *C_r_* value of the COD experienced significant negative growth.

The *C_i_* value of the Chl-*a* was 364.04 ± 39.12 mg/L, which decreased rapidly with an increasing HRT. On the 15th day, the *R*_1_ values in the PT, PE, PL, and BC groups were 93.08%, 84.61%, 82.71%, and 68.60%, respectively. On the 60th day, the *R*_1_ values of the Chl-*a* in the three plant groups were all above 90%—significantly higher than that in group BC (80.72%) (*p* < 0.05). The results indicated that the plants played an important role in the removal of Chl-*a* in the wastewater.

The *C_i_* value of the COD was 103.6 ± 3.4 mg/L. On the 15th day, the COD concentration in the PL and PT groups decreased rapidly, and the *R*_1_ values were 72.40% and 75.59%, respectively, which were significantly greater than those in the PE (46.38%) and BC (−18.41%) groups. This result indicated that the aquatic plants could effectively reduce the COD value of wastewater; the two native aquatic plants offered greater advantages than the *E. crassipes* in COD purification. Thereafter, the *R*_1_ values of the COD in the PL and PT groups remained relatively stable, while those in the PE and BC groups fluctuated significantly, indicating that the strong plant growth was conducive to the water stability. On the 60th day, the *R*_1_ values in the three plant groups were close to 70%, which was significantly higher than that in the BC group (−33%) (*p* < 0.05).

## 4. Discussion

### 4.1. Visual Observations

In this study, the native aquatic plants both grew well, but the exotic plant *E. crassipes* decayed and died. The *R*_1_ values of the TN and TP in the PE group were −3.50% and −7.40%, respectively, which aggravated water pollution. Many researchers have reported the growth restriction and death phenomena of *E. crassipes* in animal wastewater [20,21]. However, the phenomenon that stems, leaves, and roots of *E. crassipes* gradually whitened, rotted, and eventually died in the wastewater was first observed and reported in the present study, which deserves further investigation.

Sooknah and Wilkie [21] suggested that high salinity was the principal reason for the inhibited phenomenon of *E. crassipes*. Haller et al. [20] reported that seawater with salt concentrations of 2500 mg/kg had toxic effects on *E. crassipes*, which was equivalent to a conductivity of 4040 μS/cm, using a conversion factor of 1000 mg/kg = 1616 μS/cm. But the highest EC value in the present study was just 434 μS/cm.

Gerendás et al. [22] reported that numerous aquatic plants exhibit reduced growth under strict NH_4_^+^ nutrition, and develop NH_4_^+^ toxicity symptoms, including chlorosis of the leaves, overall growth suppression, and reduced root-to-shoot ratios. Ammonium levels of 188 mg/L were considered high N levels in constructed wetland cells [23]. Moreover, the unionized form of ammonia, namely, NH_3_, may contribute to the death of plants, the concentration of which depends on the ammonium ions, NH_4_^+^, and pH [24]. Reddy et al. [25] pointed out that the content of NH_3_ in wastewater is negligible if the pH is below 8.0. In the present study, the highest concentration of NH_4_^+^–N in all treatments was below 8.00 mg/L, and the pH was between 7.00 and 8.00 in the *E. crassipes* group, indicating that NH_3_ and NH_4_^+^ were not the principal reasons for the death.

Marschner [26] reported that a surplus uptake of NO_3_^−^ can be stored in the vacuoles of plant cells without harmful effects. The change trend of the NO_3_^−^–N in the PE group was totally different from the others, but normal. Its concentration increased to 9.85 mg/L from below 0.1 mg/L, which was similar to that in the study of Sooknah and Wilkie [21], and suggested that a certain amount of nitrification occurred in the PE group. Moreover, the NO_2_^−^–N increased to 6.82 mg/L, nearly 34,764 times the initial value. It is worth investigating whether this was the reason for the reduced growth and death of the *E. crassipes*. Furthermore, Wan et al. [27] demonstrated that the lethal TN and TP concentrations for *E. crassipes* were 1514.26 mg/L and 200.4 mg/L, respectively, substantially higher values than those in our experiment. Because a wide range of soluble organic compounds has been reported to be toxic to plants [28], the nature of the uncharacterized organic matter in the wastewater could be another explanation for the death in this study [29].

Plant invaders can significantly diminish the abundance or survival of native species, and may completely alter the native ecosystem in terrestrial and freshwater habitats [30]. As an invasive aquatic plant that is commonly used in water restoration engineering, the ecological risks of *E. crassipes* require further assessment. The majority of experimental studies on the effects of plant purification have been carried out in manually disposed wastewater, which differs from an in-situ water body. Moreover, as the water environment is relatively complex, when selecting aquatic plants in engineering practice, it is necessary to conduct a pre-test on the water body to be repaired, consult the literature, and draw lessons from previous experience, which can prevent secondary pollution caused by plant growth discomfort and large-scale death to a certain extent.

### 4.2. Nutrient Concentration and Absorption Capacity of Plants

It has been established that the N and P concentrations in aquatic plant tissue gradually decrease during the active growth period, while the nutrients begin to accumulate after the active growth period [31]. In this study, the TN and TP concentrations in the *L. adscendens* and *T. natans* plant tissues all decreased continuously with an increasing HRT, which was related to the rapid reduction in the inorganic N concentration in the wastewater and the rapid growth of the plant biomass in the earlier period, resulting in the decrease of nutrient elements per unit of biomass, which was similar to the findings of Debusk et al. [29].

During the process of purifying eutrophic water by means of aquatic plants, the pollutants in the water can be transferred to the plants through plant absorption, and then removed from the water by harvesting the plants. Numerous studies have demonstrated that the absorption of nutrients by aquatic plants contributes little to the removal of nutrients in wastewater, at only approximately 2% to 6% [32,33]. However, Jiang et al. [34] reported that the uptake of N and P by 17 plants accounted for 46.8% and 51.0% of the total removal of water pollutants, respectively. In this study, the absorption contribution rates (*R*_2_) of the TN and TP in the two native plants were between 34.5% and 43.5%, suggesting that *R*_2_ was affected by the plant species and water pollution degree. As reported by Brix [35] and Peterson and Teal [36], the uptake of N and P by plants contributes significantly to the removal of nutrients in low-load constructed wetlands, and plays a limited role in high-load systems. Therefore, it is ineffective to evaluate the absorptive capacity of plants without considering the research background.

Jiang et al. [34] demonstrated that suitable water purification plants can be selected directly by means of the biomass index. In this study, more rapid plant biomass growth resulted in a superior purification effect of the water quality. Within 60 days, the biomass of *L. adscendens* increased 5.4 times, and the removal rates of the TN and TP in the water body were higher than 87.0%. This indicated that, with a large biomass and rapid growth capacity, the native aquatic plants *L. adscendens* and *T. natans* can be favored in selection for the relevant engineering practice of water ecological restoration.

### 4.3. Changes in pH, DO, ORP, and EC in Wastewater

pH is an important factor affecting nutrient removal; excessive acidity and alkalinity of the water body may aggravate the release of N and P in the sediment, which is the smallest when the water pH is neutral [37,38]. Moreover, a change in the water pH will affect the growth of cyanobacteria; a higher pH can promote the growth of algae cells, while a lower pH has the opposite effect [39]. In this study, the pH values in the PL and PT groups were reduced to neutral, which indicates that planting *L. adscendens* and *T. natans* in an alkaline environment can effectively reduce the pH value of the water body, and subsequently inhibit the release of N and P nutrients in the sediment and the growth of algae.

In this study, the phenomenon whereby that the DO concentration in the PT group was relatively low compared to that of other groups could be explained by the reduced oxygen diffusion from the atmosphere into the water column, owing to the plant cover, higher root respiration rates, and oxygen uptake by the microorganisms attached to the roots [21]. Therefore, for effective water purification, the grown biomass of aquatic macrophytes must be removed from water bodies to maintain an optimal plant density and permit increased oxygen exchange.

The EC represents the ability of a solution to conduct current, and indirectly infers the total ion concentration in the water. In this experiment, the EC values in the PL, PT, and BC groups exhibited a downward trend with the removal of N and P in the wastewater. However, the EC value in the PE group did not exhibit the same downward trend with the relatively fast removal rate of the nutrients, which could be owed to the growth stopping, the color change from purple to white, and the gradual decay of its roots. It has been established that the purification effects of aquatic plants on wastewater are closely related to their root systems. The allelochemicals secreted by the huge root systems of plants may influence the water’s EC values, which requires further study [40].

### 4.4. Nutrient Removal Efficiency by Aquatic Plants

In the process of using aquatic plants to repair eutrophic water bodies, the coupling effects of the microorganisms and plants play an important role in nutrient removal from the wastewater. The roots of floating aquatic plants provide substrates for microbial communities and aerobic microsites in a generally anaerobic environment [41]. Microbial communities promote nutrient assimilation by plant roots and largely aid in chemical transformations, including nitrification and denitrification [36,42,43].

N has a complex biogeochemical cycle, with multiple biotic/abiotic transformations involving seven valence states (+5 to −3) [14]. Effective means of N removal in wastewater include assimilation and absorption by plants, volatilization of NH_3_, nitrification/denitrification, entrapment of particulate matter (organic nitrogen) by the extensive root systems, and settling [21]. In this study, the rapid increases in the NO_3_^−^–N and NO_2_^−^–N in the PE group indicated that nitrification occurred extensively, whereby NH_4_^+^–N was oxidized into NO_2_^−^–N and NO_3_^−^–N by nitrifying bacteria, while the reduction in the NO_3_^−^–N and NO_2_^−^–N could be owed to the plant uptake and denitrification. The NH_4_^+^ in wastewater may be lost from the system through volatilization, taken up by plants and microbes, or oxidized into nitrate during the nitrification process. Kronzucker et al. [44] reported that NH_4_^+^ is the predominant form of inorganic N available for plant uptake. In the beginning of the present study, the removal of NH_4_^+^ in the plant groups was significantly higher than that in the BC group, indicating that the aquatic plants played a significant role in the removal of NH_4_^+^. Reddy et al. [25] pointed out that losses of NH_3_ through volatilization are insignificant if the pH value is below 7.5, and the losses are very often not serious if the pH is below 8.0. This suggests that the NH_4_^+^–N removal in the plant cultures may have been primarily owed to the plant uptake and nitrification, along with a lesser level of volatilization, given that these systems had less alkaline pH in this study.

The P cycle is fundamentally different from the N cycle [14]. P cannot leave the water through gaseous volatilization, and the majority of P exists in the form of insoluble phosphate [45]. In this study, the TP removal rates of the two native plants in the wastewater were between 88.72% and 95.90%, which were substantially higher than those of the blank control (38.32%) and *E. crassipes* (26.82%) groups. The TP absorption contribution rates of the native plants were between 34.5% and 43.5%. The results indicated that plants play an important role in the removal of TP in wastewater; the uptake of aquatic plants contributes significantly to the reduction; and the other processes, such as desorption, precipitation, dissolution, microbial uptake, fragmentation, leaching, mineralization, sedimentation, and burial, work together.

In this study, a major part of the degradation of pollutants (COD) in the wastewater could be attributed to the microorganisms around the roots, which may establish a symbiotic relationship with the plants. As an important indicator for evaluating the eutrophication level, Chl-*a* is significantly corrected with the biomass of phytoplankton. While competing with algae for nutrients and living space, many aquatic plants can produce allelochemicals, which are considered to play a role in regulating the distribution of the phytoplankton population, and may become an important means of controlling algal blooms [46]. In this study, the *R*_1_ values of the Chl-*a* in the three plant groups were all significantly higher than that in the BC group, indicating that the aquatic plants could inhibit the growth of algae in the wastewater.

Overall, the existence of aquatic plants can enrich the water biodiversity, improve the stability of the aquatic ecosystem resistance, improve the quality of the water environment, and promote the removal of N and P by other factors. For example, the oxygen secretion function of aquatic plant roots can promote the growth and metabolism of nitrifying bacteria, denitrifying bacteria and other rhizosphere microorganisms, and accelerate the decomposition of pollutants. The introduction of appropriate aquatic plants into eutrophic water bodies may facilitate long-term improvement in the water quality.

## 5. Conclusions

Floating aquatic plants can play a significant role in purifying eutrophic water. The present results show that *L. adscendens* and *T. natans* are both promising native aquatic plants to be applied to the restoration of swine manure wastewater; plants can adjust the pH of the wastewater, improve the DO and ORP, reduce the EC value and COD, and inhibit the growth of algae; the main removal pathways of N and P include the plant uptake and the synergistic effects of plant roots and microorganisms, revealing that the suitable water purification plants can be selected directly by means of the large biomass, rapid growth capacity, and developed root system index; for effective water purification, the grown biomass of aquatic macrophytes must be removed from water bodies to maintain an optimal plant density and permit increased oxygen exchange; as an invasive aquatic plant that is commonly used in water restoration engineering, the ecological risks of *E. crassipes* require further assessment. In conclusion, our results have indicated that the native aquatic plants have great potential in the ecological restoration of water bodies.

## Figures and Tables

**Figure 1 ijerph-17-01103-f001:**
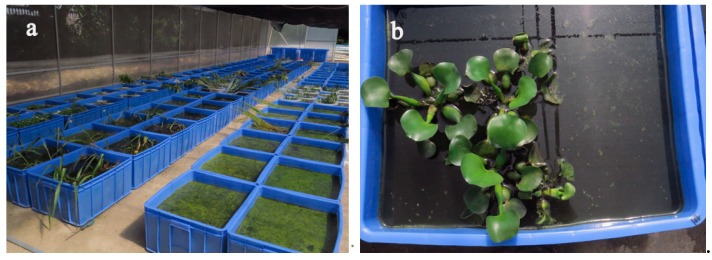
Experimental scene on the first day: (**a**) all plants and (**b**) *Eichhornia crassipes*.

**Figure 2 ijerph-17-01103-f002:**
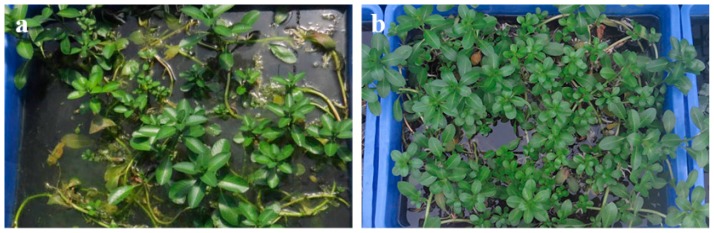
Growth status of *Ludwigia adscendens* in swine manure wastewater: (**a**) day 0 and (**b**) day 15.

**Figure 3 ijerph-17-01103-f003:**
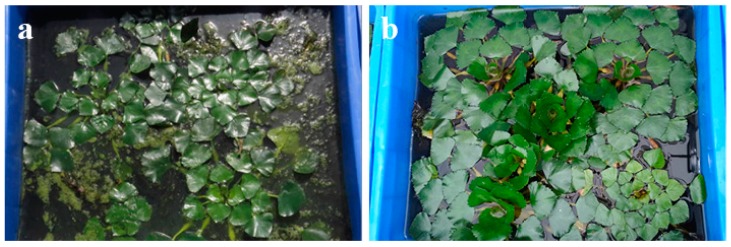
Growth status of *Trapa natans* in swine manure wastewater: (**a**) day 0 and (**b**) day 15.

**Figure 4 ijerph-17-01103-f004:**
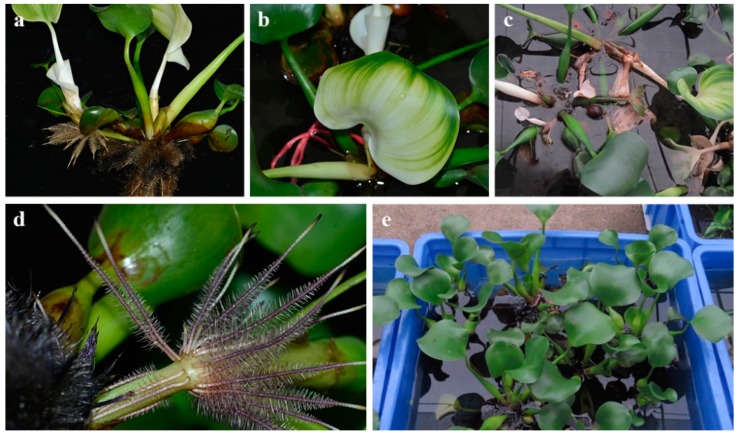
Growth status of *E. crassipes* in two types of eutrophic water bodies: (**a**) to (**c**): the roots and leaves turned white and rotted gradually (in the swine manure wastewater); and (**d**) to (**e**) the roots were purple and grew vigorously (in the river).

**Figure 5 ijerph-17-01103-f005:**
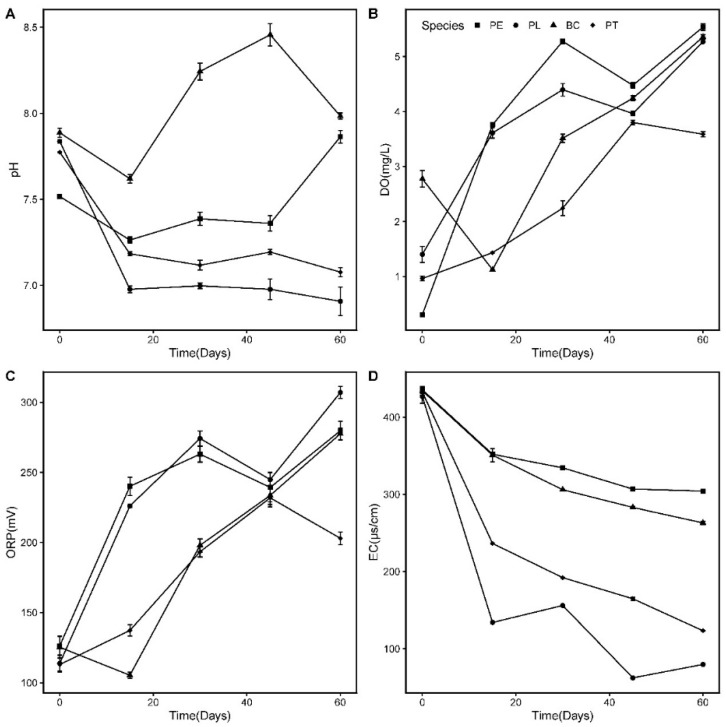
Changes in physical and chemical indicators in wastewater: (PE) *E. crassipes*; (PL) *L. adscendens*; (BC) *black control*; (PT) *T. natans* (the same in the following figures). (**A**): the value of pH; (**B**): the concentration of DO; (**C**): the value of ORP; (**D**): the value of EC.

**Figure 6 ijerph-17-01103-f006:**
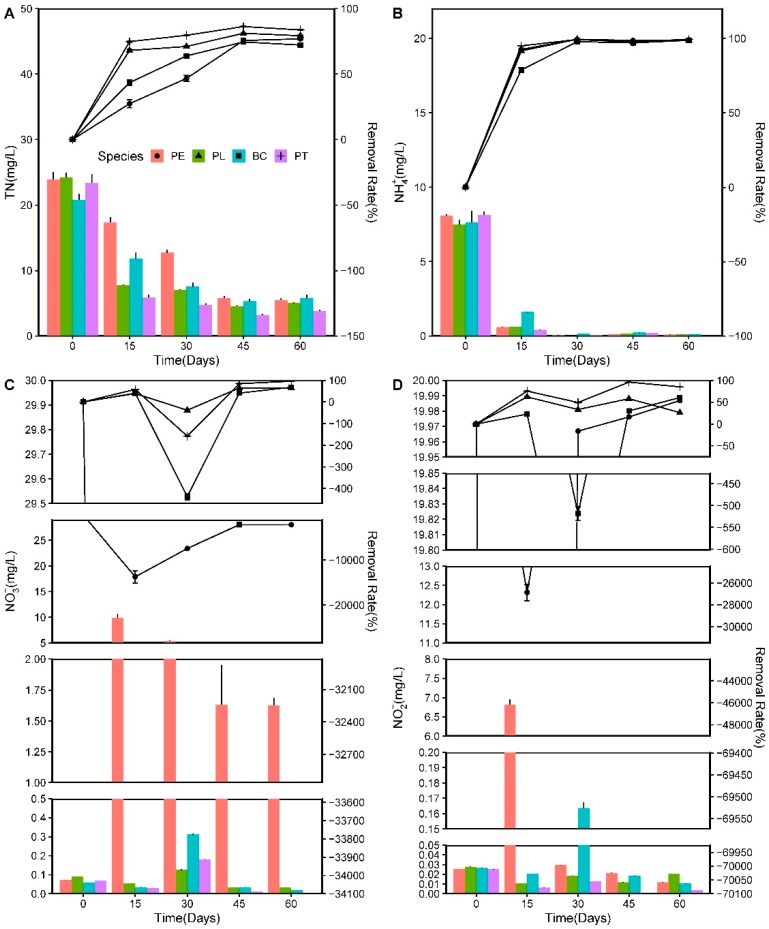
Variations in N concentration and homologous removal rate in the experiment: (**A**) concentration and removal rate of TN; (**B**) concentration and removal rate of NH_4_^+^; (**C**) concentration and removal rate of NO_3_^−^; and (**D**) concentration and removal rate of NO_2_^−^.

**Figure 7 ijerph-17-01103-f007:**
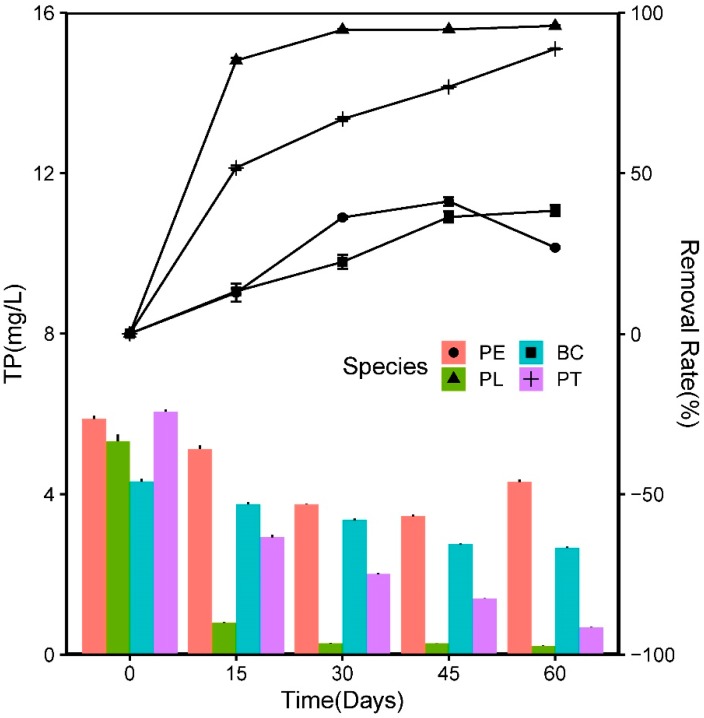
Variations in TP concentration and the homologous removal rate during experiment.

**Figure 8 ijerph-17-01103-f008:**
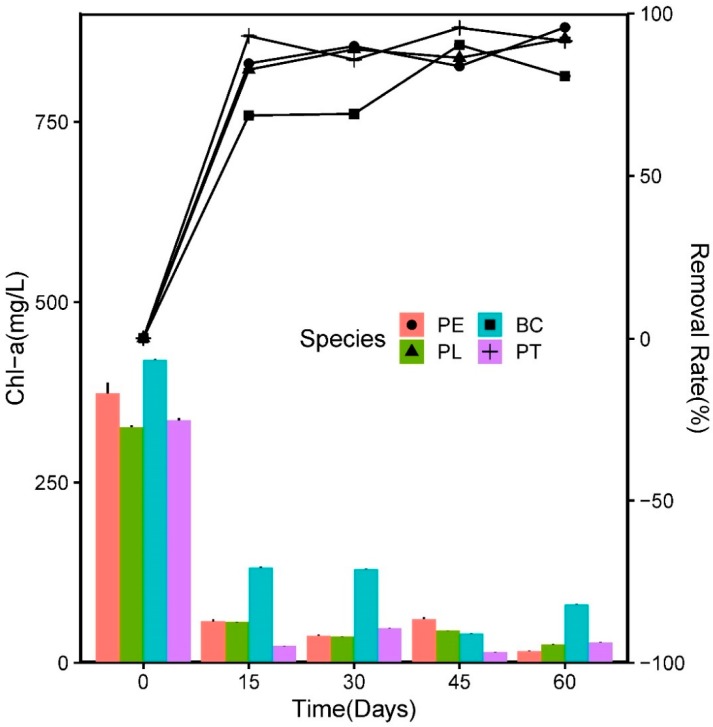
Variations in Chl-a concentration and the homologous removal rate in the experiment.

**Figure 9 ijerph-17-01103-f009:**
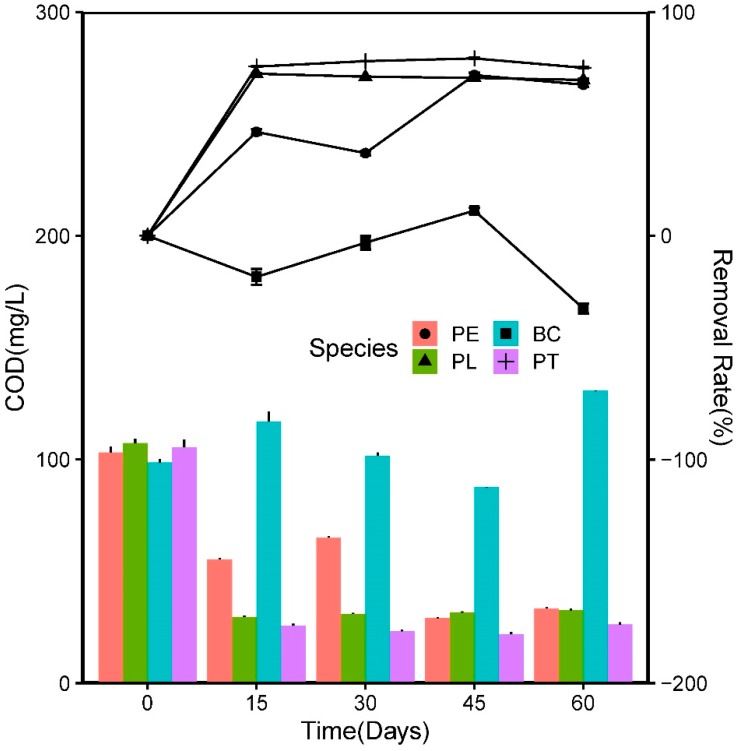
Variations in chemical oxygen demand (COD) concentration and the homologous removal rate in the experiment.

**Table 1 ijerph-17-01103-t001:** Biomasses of floating aquatic macrophytes after 60 days of treatment in swine manure wastewater (g/pot, fresh weight).

Treatment	Initial Fresh Weight	Harvest				Net Biomass Growth Rates
		1	2	3	Total	
*E. crassipes*	200.0	100.1	100.4		200.5	0.2%
*T. natans*	200.0	91.6	103.5	776.7	971.8	385.9%
*L. adscendens*	200.0	423.3	400.2	456.2	1279.7	539.8%

Note: Harvest 1 was on the 15th day of the experiment, Harvest 2 was on the 30th day, and Harvest 3 was on the 60th day (the same applies below).

**Table 2 ijerph-17-01103-t002:** Total nitrogen (TN) concentration and total uptake of floating aquatic macrophytes after 60 days of treatment in swine manure wastewater (mg/g, dry weight).

Treatment	TN Content (mg/g, Dry Weight)							
	Initial			1st Harvest	2nd Harvest	3rd Harvest		
	Roots	Shoots	Fruit	Shoots	Shoots	Roots	Shoots	Fruit
*T. natans*	15.81	31.85	10.00	22.30	15.62	11.23	11.50	7.06
*L. adscendens*		31.42		18.23	11.30	10.56	6.32	

**Table 3 ijerph-17-01103-t003:** Total phosphorus (TP) concentration and total uptake of floating aquatic macrophytes after 60 days of treatment in swine manure wastewater.

Treatment	TP Content (mg/g, Dry Weight)
Initial	1st harvest	2nd harvest	3rd harvest
Roots	Shoot	Fruit	Shoots	Shoots	Roots	Shoots	Fruit
*T. natans*	2.15	2.99	3.80	2.80	2.64	1.88	2.45	3.22
*L. adscendens*		3.10		2.90	2.40	2.36	1.85

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
