# Peer review of "Assessment of the Nutrient Removal Potential of Floating Native and Exotic Aquatic Macrophytes Cultured in Swine Manure Wastewater"

_ijerph, 2020, doi:10.3390/ijerph17031103_

Round 1
Reviewer 1 Report
Manuscript ijerph-704482 reports a well-done study and is of great interest. Manuscript has a correct structure and includes clear descriptions and explanations of the experimental design. The manuscript has valuable data. It presents data in an easy to follow way. The analytical approach is appropriate and the conclusions are coherent with the data and give some perspective for future studies. Following comments aim to improve the quality of the manuscript:
- Manuscript has some grammar and language issues, which need to be addressed. Please, careful review of the text.
- Please add a schematic representation of the experimental setup
- Please, increase the readability of figures 4 and 5
Reviewer 2 Report
The paper describes the application of floating aquatic plants to reduce nitrogen pollution, from polluted sources, to reduce nitrogen-compound emissions. The results, in this manuscript, point out the component that is one of the most efficient ways in the agricultural point and/or non-point source pollution ecosystem. As the data confirm your concerning for that nitrogen contamination might enter water bodies through organic matters through the aquatic ecosystem, this manuscript may be indicated the risk for adoption of a natural treatment system like the system presented here in a closed system, beyond an open system. I find your work interesting and the methodology is well explained. However, the authors need to modify the introduction part and highlight the novelty, as well as the objective of the study. There is one specific question: How the authors get the driving force from your agricultural sources in response to nitrogen contamination containing about the removing efficiency in an ecosystem? Why do you want to study polluted agricultural dirt/manure in water bodies? Why this area? And why these two species? You need more explanation for our international readers for the following questions. Some details are redundant and can be summarized in your paper reviews. Please revise the document according to the following suggestions:
In the introduction section the literature should be increased for the theory of stated preference. What is your hypotheses you want to detect from 12 rectangular plastic tanks at Sihui, Guangdong Province, South China? There is no hypothesis about the comparison of locations. Please indicate. The authors do not present a comparative table of all the hypothesis statements and the statistical findings in an entire scope. If the removal efficiencies were detected very high, what levels, beyond water table depth, did you detect in your findings to design for a maximum in efficiency? Please summarize. In your section of 3.1., please keep the same font style for the typing words between your parentheses. Please note a superscript (4) in your name. Who is belonging to University of Chinese Academy of Sciences, Beijing 10049, China? I cannot see any persons’ name(s) detected by the superscript (4).Author Response
Please see the attachment.

Round 2
Reviewer 2 Report
All questions have been cleared.